# Co-crystallization of atomically precise metal nanoparticles driven by magic atomic and electronic shells

Juanzhu Yan[1], Sami Malola[2], Chengyi Hu[1], Jian Peng[1], Birger Dittrich[3], Boon K. Teo[1], Hannu Häkkinen [2], Lansun Zheng[1] & Nanfeng Zheng [1]

This paper reports co-crystallization of two atomically precise, different-size ligand-stabilized nanoclusters, a spherical $(AuAg)_{267}(SR)_{80}$ and a smaller trigonal-prismatic $(AuAg)_{45}(SR)_{27}(PPh_3)_6$ in 1:1 ratio, characterized fully by X-ray crystallographic analysis $(SR = 2,4\text{-}SPhMe_2)$. The larger cluster has a four concentric-shell icosahedral structure of $Ag@M_{12}@M_{42}@M_{92}@Ag_{120}(SR)_{80}$ $(M = Au \text{ or } Ag)$ with the inner-core $M_{147}$ icosahedron observed here for metal nanoparticles. The cluster has an open electron shell of 187 delocalized electrons, fully metallic, plasmonic behavior, and a zero HOMO-LUMO energy gap. The smaller cluster has an 18-electron shell closing, a notable HOMO-LUMO energy gap and a molecule-like optical spectrum. This is the first direct demonstration of the simultaneous presence of competing effects (closing of atom vs. electron shells) in nanocluster synthesis and growth, working together to form a co-crystal of different-sized clusters. This observation suggests a strategy that may be helpful in the design of other nanocluster systems via co-crystallization.

[1] State Key Laboratory for Physical Chemistry of Solid Surfaces, Collaborative Innovation Center of Chemistry for Energy Materials, and National & Local Joint Engineering Research Center for Preparation Technology of Nanomaterials, College of Chemistry and Chemical Engineering, Xiamen University, 361005 Xiamen, China. [2] Departments of Physics and Chemistry, Nanoscience Center, University of Jyväskylä, FI-40014 Jyväskylä, Finland. [3] Heinrich-Heine Universität Düsseldorf, Anorganische Chemie und Strukturchemie, Universitätsstrasse 1, Gebäude 26.42.01.21, 40225 Düsseldorf, Germany. Correspondence and requests for materials should be addressed to H.Ḧk. (email: hannu.j.hakkinen@jyu.fi) or to N.Z. (email: nfzheng@xmu.edu.cn)

A tomically precise metal nanoparticles (also called metal nanoclusters) stabilized by organic ligands are of interest due to their intermediate size that bridges atoms and bulk solids, inducing distinct physico-chemical properties in the quantum-size regime[1–4]. In recent years, there has been explosive advances in building various nanoarchitectures via a wide range of synthetic strategies[5]. However, factors driving the thermodynamic stability of discrete nanometer sizes are largely unknown[6,7].

Ligand-stabilized metal nanoclusters are typically produced in solution by reducing metal salts in the presence of protecting ligands. Achieving a high degree of control over their size and atomic composition is generally a complex process that is affected by several factors during synthesis such as the metal: ligand ratio, reaction temperature, rate of reduction, post-processing methods such as size focusing, and purification. A general view is that the formation of the protected clusters can be stopped in the process of either nucleation or etching once a magic size and composition is reached[3]. These magic sizes and compositions are then entities trapped in local free-energy minima surrounded by kinetic barriers in the multi-dimensional phase space. Three factors are commonly considered to play important roles in the formation of magic cluster sizes and compositions: (i) Favorable surface chemical structure (the ability to protect the metal core from the environment); (ii) Favorable atomic packing, often seen as spherical structures of concentric polyhedral shells; (iii) Closed-shell electronic structure stabilizing smaller, molecule-like nanoclusters through opening of a large energy gap between the highest occupied molecular orbital (HOMO) and the lowest unoccupied molecular orbital (LUMO), the so-called HOMO-LUMO gap. The process has thus certain analogies to the formation of magic metal nanoclusters in gas-phase experiments, where the crucial roles of electronic shells and atomic packing (atomic shells) have been discussed extensively since the 1980s[8–12].

In this work, we present and analyze direct evidence that the formation of both atomically closed-shell and electronically closed-shell nanoclusters is possible concurrently in a synthesis that produces atomically closed-shell intermetallic $(AuAg)_{267}(2,4\text{-}SPhMe_2)_{80}$ nanoparticles $(AuAg)_{267}$ and electronically closed-shell $(AuAg)_{45}(2,4\text{-}SPhMe_2)_{27}(PPh_3)_6$ nanoclusters $(AuAg)_{45}$. These species were identified from a co-crystal having an unusual packing motif for metal nanoparticles. The co-crystallization of two such different-sized nanoparticles, stabilized by competing mechanisms (atomic vs. electronic shell effects) to achieve magic sizes and compositions suggests a valid strategy that might be helpful in the synthesis and co-crystallization of other bimodal nanocluster systems.

## Results and Discussion

**Synthesis and characterization**. The mixture of $(AuAg)_{267}$ and $(AuAg)_{45}$ nanoclusters was prepared by reducing the metal precursors $AuPPh_3Cl$ and $AgNO_3$ (1:1 ratio) with $NaBH_4$ in the presence of 2,4-dimethylbenzenethiol (HSR), triphenyphosphine ($PPh_3$), tetraphenylphosphonium bromide ($PPh_4Br$), and triethylamine in methanol/dichloromethane at 0 °C (see SI for more details). Transmission electron microscopic (TEM) analysis revealed that the as-prepared Au–Ag compounds were dominantly in two sizes, nearly 50/50, for ~2.48 nm and ~1.10 nm (Supplementary Fig. 1). To determine their detailed molecular structure, much effort was devoted to crystallize the as-prepared products into dark single crystals (Supplementary Fig. 2) suitable for X-ray diffraction by slowly diffusing hexane into the dichloromethane solution at 0 °C. The X-ray single-crystal structure analysis revealed the co-crystallization of both

$(AuAg)_{267}$ and $(AuAg)_{45}$ nanoclusters (Fig. 1a). The $(AuAg)_{267}$ nanoparticle is spherically shaped and the $(AuAg)_{45}$ nanocluster is a trigonal prism. These two differently shaped nanoclusters are hierarchically assembled in a hexagonal $P6_3/m$ space group in a 1:1 ratio (Fig. 1b, Supplementary Figs. 3 and 4).

**Atomic structure**. A detailed analysis revealed that $(AuAg)_{267}$ nanoparticle exhibits a remarkable high-symmetry three-layer Mackay icosahedron (MIC) molecular structure with efficient *fcc* atom packing (Fig. 2a, b), with the fourth layer in anti-Mackay configuration. The geometrical anatomy of the $(AuAg)_{267}$ nanoparticle can be represented by a four-shell metal structure $v_0(1)@v_1(12)@v_2(42)@v_3(92)@w_4(120)$, which is protected by a buckyball-like $S_{80}$ thiolate-ligand shell. This "shell-by-shell" or "Matryoshka doll" representation is depicted in Fig. 2c–f. The 120 atoms in the fourth shell are mutually connected to form a semiregular polyhedron containing 20 $v_2$ triangles, 60 squares, and 12 pentagons. The center of each $v_2$ triangular and square face in the fourth shell is capped by a thiolated ligand (Fig. 2g, h), such that the 80 sulfur atoms from the thiolated ligands define a slightly distorted buckyball (Fig. 2i). A more detailed description of the $(AuAg)_{267}$ structure is found in the SI text (Supplementary Fig. 5 and Supplementary Table 3). The $(AuAg)_{45}$ cluster has a trigonal prismatic shape, containing a two-shell tri-capped-prism metal framework and protected by 27 thiolates and 6 triphenylphosphines (see Supplementary Discussion, Supplementary Figs. 6 and 7). The molecular structure is almost identical to $Au_9Ag_{36}(SPhCl_2)_{27}(PPh_3)_6$ reported by us previously[13].

Mackay introduced the concept of the hierarchical icosahedral structure 50 years ago[14]. The cluster of 55 atoms within the second icosahedral shell occurs frequently, as determined by mass spectroscopy or theoretically forecasted, and thus was often referred to as Mackay icosahedron. In the past decades, the two-layer $M_{55}$ Mackay icosahedra[14] have also been disclosed in a diversity of nanosized noble metal clusters via single-crystal x-ray diffraction, namely, a three-shell 145-metal-atom $Pd_{145}(CO)_x(\text{-}PEt_3)_{30}$ ($x \approx 60$) cluster[15], a Pd-Pt four-shell 165-metal-atom $(\mu_{12}\text{-}Pt)Pd_{164-x}Pt_x(CO)_{72}(PPh_3)_{20}$ ($x \approx 7$) cluster[16] and 133-metal-atom $Au_{133}(SR)_{52}$ one[17,18], as well as a two-shell 55-metal-atom $Pd_{55}(^{Pi}Pr_3)_{12}(\mu_3\text{-}CO)_{20}$ cluster[19], which were of particular significance as revelatory for predicting the structures of these air-sensitive 55-metal-atom $M_{55}L_{12}Cl_x$ (M = Au, Pd, Rh, or Ru; x = 6 or 20) clusters reported by Schmid et al.[20,21]. In the present work, the next Mackay overlayer based on icosahedral $M_{55}$ is crystallographically observed in $(AuAg)_{267}$, giving rise to well-known three complete Mackay metal shells of 147 atoms[14], $v_0(1)@v_1(12)@v_2(42)@v_3(92)$, at variance to embryonic growth of decahedral ($Au_{102}$[22], $Au_{130}$[23], $Au_{246}$[24], $Ag_{136}$, and $Ag_{374}$[25]) and *fcc* cuboctahedral $Au_{279}$[26] clusters.

**Distribution of Au and Ag**. Considering the positional disorder of metal atoms might occur in the kernels of $(AuAg)_{267}$ nanoparticle and $(AuAg)_{45}$ nanoclusters, we have used different methods to get insight into their metallic distributions. The outermost metallic shells of $(AuAg)_{267}\cdot(AuAg)_{45}$ co-crystal consist of Ag atoms which are in sync with the coordination modes of Ag-SR and $Ph_3P$-Ag-SR in ways analogous to previous works[27–30], but at odds with the typical $Au_n(SR)_{n+1}$ ($n = 1$ or 2) staple motifs commonly observed in many thiolated gold clusters[22,31,32]. As shown in Table 1, the approximate composition of the molecular formula in term of central Au (Ag) and Ag (Au) atoms in the inner shells (Au atoms for the $(AuAg)_{45}$ cluster and Ag for the $(AuAg)_{267}$ nanoparticle) was established from $Au_xAg_{1-x}$ occupancies based upon least-squares refinement of the X-ray data. The total crystallographically estimated contribution

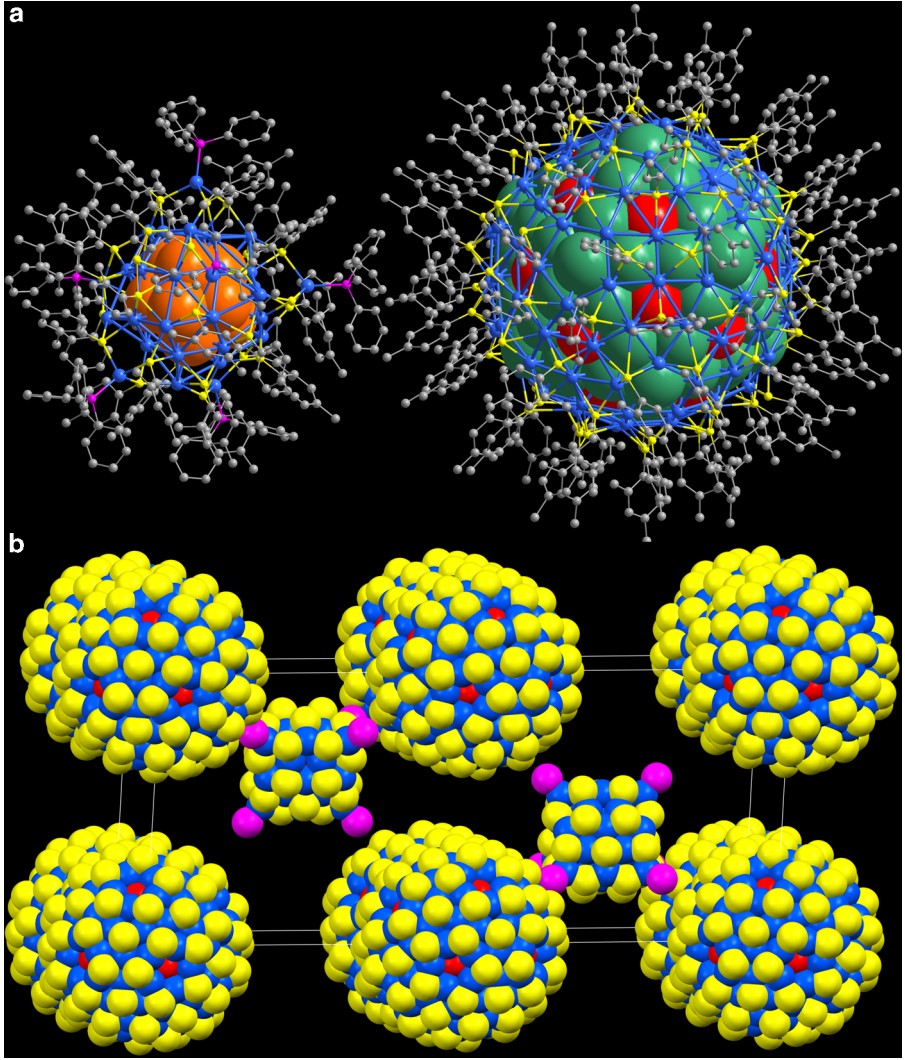

**Fig. 1** Cocrystal structure of $(AuAg)_{267} \bullet (AuAg)_{45}$. **a** The total structure of the plasmonic $(AuAg)_{267}$ nanoparticle co-crystallized with the molecule-like $(AuAg)_{45}$ cluster. All hydrogen atoms are omitted for clarity. **b** Unit cell of the three-dimensional structure of spherical $(AuAg)_{267}$ nanoparticles and trigonal prismatic $(AuAg)_{45}$ nanoclusters in space-filling model. All hydrogen and carbon atoms are omitted for clarity. Color code: orange Au, sea green and red AuAg, blue Ag, yellow S, magenta P, gray C

of $x = 93.8$ for $Au_x Ag_{312-x}$ (2.33:1 molar ratio of Ag/Au for total 312 metallic atoms in $(AuAg)_{267} \cdot (AuAg)_{45}$) tallied with that found from an inductively coupled plasma mass spectrometric (ICP-MS) Ag/Au determination (2.36:1 molar ratio of Ag/Au for a total of 312 metallic atoms in $(AuAg)_{267} \cdot (AuAg)_{45}$), as well as the outcome of an energy dispersive X-ray spectroscopic (EDS) analysis (2.36:1 molar ratio of Ag/Au for total 312 metallic atoms in $(AuAg)_{267} \cdot (AuAg)_{45}$). Interestingly, there are some regularities in the Au–Ag distribution in $(AuAg)_{267} \cdot$ and $(AuAg)_{45}$ as described below.

For $(AuAg)_{267}$, the central site has 100% Ag occupancy, and all sites on $C_3$ axes of the 147-atom icosahedral kernel are Ag-rich (Ag occupancies >85%), and the remaining sites of the first ($v_1$) and second ($v_2$) shells are disordered with roughly 50-50 occupancies of Au and Ag. The same segregation phenomenon occurs in the third ($v_3$) shell wherein the Ag-rich (90–100%) atoms on 3-fold axes are surrounded by Au-rich (70–85%) atoms on the vertices and the edges. Despite these seemingly random disorder (note that Au and Ag form completely disordered solid solutions), a careful examination of the relative Au:Ag occupancies revealed some regularities, as follows. The first three shells, along with the central atom ($v_0$) and the anti-Mackay-like surface

shell ($\omega_4$), exhibit alternating Ag- and Au-rich shells (average occupancies in parentheses): $v_0$(100% Ag)@$v_1$(ca. 60% Au) @$v_2$(ca. 60% Ag)@$v_3$(ca. 80% Au; ca. 90% Ag on 3-fold axes) @$\omega_4$(100%Ag) (see Supplementary Fig. 8 and CCDC-1839942). This alternating shell behavior maximizes the Au–Ag heteronuclear bonds at the expenses of homonuclear Au–Au and Ag–Ag bonds due to the disparity in the electronegativity of Au and Ag, which increases the bond energy due to the ionic character of the heteronuclear bonds[33]. DFT calculated results also suggested that Ag is systematically positively charged and Au negatively charged in $(AuAg)_{267}$ (Supplementary Table 4), so there is always a slight electron transfer from Ag to Au. Unlike $(AuAg)_{267}$, the 9 atoms in $M_9$ tricapped trigonal prism core of the $(AuAg)_{45}$ cluster, are simply composed Au with occupancies nearly 100% (see CCDC-1839941).

**Interparticle assembly**. Apart from the self-organized process for the formation of individual nanoparticles at the atomic level, $(AuAg)_{267}$ and $(AuAg)_{45}$ components serve as building blocks, hierarchically assembled into a three-dimensional structure with a relative arrangement similar to PtB or anti-NiAs structures (Fig. 1b and Supplementary Figs. 9–11). In the crystal lattice, each

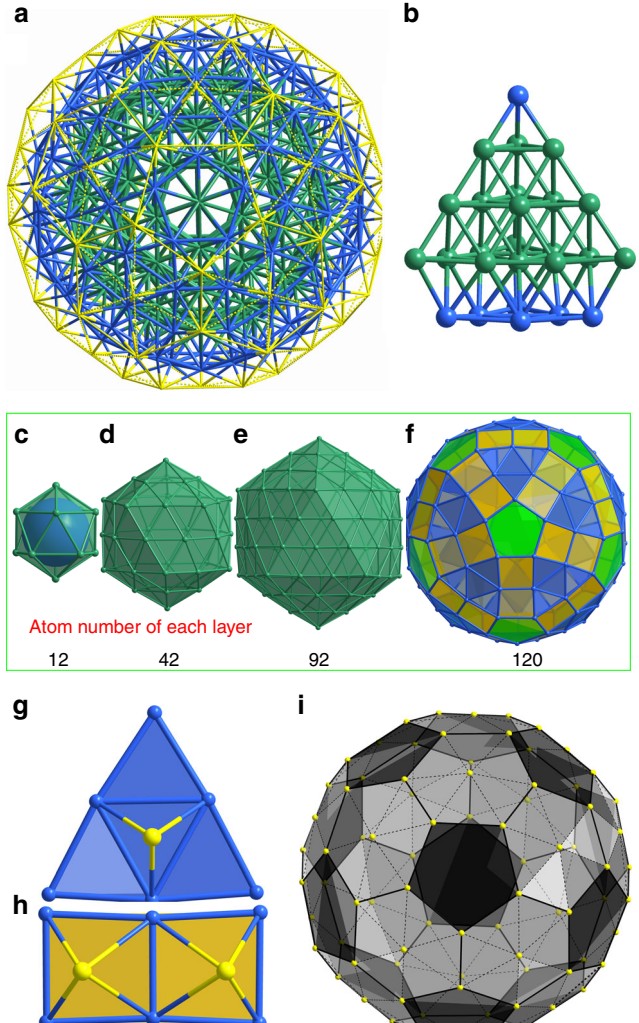

| Table 1 Au/Ag ratio in (AuAg)$_{267}$·(AuAg)$_{45}$ characterized by various methods | | | | |
|---|---|---|---|---|
| **Method** | **Ag/107** | **Au/197** | **Molar (Ag)** | **Molar (Au)** | **Ag:Au molar ratio** |
| ICP-MS | 139.22 | 108.53 | 1.30 | 0.55 | 2.36:1 |
| EDS | — | — | 0.702 | 0.298 | 2.36: 1 |
| X-ray | — | — | 218.2 | 93.8 | 2.33: 1 |

**Fig. 2** Representative molecular structure of the plasmonic (AuAg)$_{267}$ nanoparticle. **a** Buckyball-shape (AuAg)$_{267}$ nanoparticle. **b** Layered ABCAC packing (or ABCB if the central Ag atom is ignored) of metal atoms. **c**–**f** Anatomy of metal framework of (AuAg)$_{267}$ nanoparticle: the first Mackay icosahedral shell ($v_1$) with 12 M atoms (**c**); the second Mackay icosahedral shell ($v_2$) with 42 M atoms (**d**); the third Mackay icosahedral shell ($v_3$) with 92 M atoms (**e**); and the fourth anti-Mackay shell ($\omega_4$) with 120 Ag atoms (**f**). **g** Three-fold coordinated thiolate centered on six-Ag-atom triangle (highlighted in blue in panels (**f**) and (**g**)). **h** Two tetradentate thiolates capped on two edge-shared squares (highlighted in yellow in panels (**f**) and (**h**)). **i** Sulfur atoms (yellow) arranged in a slightly distorted truncated icosahedron (buckyball). All hydrogen and carbon atoms are omitted for clarity. Color code: sea green and red AuAg, blue Ag, yellow S

(AuAg)$_{267}$ is surrounded by six octahedral-arranged (AuAg)$_{45}$ nanoclusters while each (AuAg)$_{45}$ is neighbored by six (AuAg)$_{267}$ nanoparticles arranged in a trigonal prismatic fashion. The particle-to-particle distance is 2.78 nm. The effective size ratio $\gamma = R_{small}/R_{large}$ of nanoparticles that are arranged in the PtB-type superlattice is usually in the range of ~0.4. However, structural analysis indicate that the sizes of (AuAg)$_{267}$ and (AuAg)$_{45}$ are 1.65 nm (Supplementary Fig. 3) and 1.05–1.25 nm (Supplementary Fig. 4), respectively ($0.636 \leq \gamma \leq 0.757$)[34]. The values 0.636–0.757 significantly exceed the size ratio predicted for a stable PtB-type structure. To show why such interactions do not lead to more closely packed CsCl or AlB$_2$-type structures that entropy alone would favor, we focused on the rigid conformation

and specific molecular shape of the particles' surface layer. As mentioned above, the (AuAg)$_{267}$ nanoparticle has a geometrically-isotropic spherical structure whose thiolated ligands are nearly homogeneously arranged on its metal core. (AuAg)$_{45}$ particle is coordinated by two different ligands which are anisotropically distributed. As shown in Fig. 3a–c, each (AuAg)$_{45}$ nanoparticle has six nearest (AuAg)$_{267}$ neighbors coincide well with the directions of the six phosphine ligands of (AuAg)$_{45}$ arranged in a trigonal prismatic fashion. The parallel thiolated ligands of (AuAg)$_{45}$ are situated on the crack between two neighboring (AuAg)$_{267}$ particles. In this way, the surface symmetry of two different nanoparticles is perfectly matched. The interacting ligands (phosphines of (AuAg)$_{45}$ and thiolates of (AuAg)$_{267}$) between diblocks are akin to "socket-plug" combination via C-H···π interactions (Fig. 3d, e). The clustering of different ligands on the periphery of diblocks leads to packing constraints and further generates the intriguing and unconventional PtB-type structure. The cocrystalline superlattice of (AuAg)$_{267}$ and (AuAg)$_{45}$ provides a system suitable for engineering hierarchical structures and also offers insights into the assembly formation mechanism at the atomic level. Both the incommensurate interactions and shape disparities are untangled during their self-organization.

**Optical properties**. Figure 4a shows the UV-vis absorption spectra for (AuAg)$_{267}$ alloy nanoparticles with (AuAg)$_{45}$ content, as well as the two corresponding independent species. Only one plasmon band at 460 nm is observed in (AuAg)$_{267}$·(AuAg)$_{45}$ CH$_2$Cl$_2$ solution. The distinct difference between the (AuAg)$_{267}$ nanoparticle and the (AuAg)$_{45}$ nanocluster was demonstrated by comparing their absorption spectra. The (AuAg)$_{45}$ nanocluster has a structured absorption spectrum with absorption bands at 434 and 570 (shoulder) nm, behaving as small molecules rather than as typical metallic nanoparticles. The visible absorption of the co-crystallized compound in CH$_2$Cl$_2$ mainly arises from the surface plasmon band of 267 Au–Ag alloyed composite. The dominance of the plasmon absorption band of (AuAg)$_{267}$ nanoparticles in the dissolved 1:1 co-crystal solution is due to the large discrepancy in extinction coefficients between the large and small clusters. Their molar absorptivity difference at 532 nm was almost a multiple of ten in CH$_2$Cl$_2$ (Supplementary Fig. 12).

**Electronic structure**. We analysed the electronic structure of both nanoparticles using the density functional theory (DFT, for details see Methods). The count of free metallic electrons in the neutral (AuAg)$_{267}$ cluster is 267–80 = 187[35]. Due to its near-spherical core shape, it is reasonable to compare the electronic structure of the metal core to the simplified spherical jellium model that describes quantization of electrons in a uniform electron gas trapped in an infinitely deep, spherical potential well[36]. This model predicts a major energy gap at 186 electrons, in a shell configuration of 1S$^2$ 1P$^6$ 1D$^{10}$ 2S$^2$ 1F$^{14}$ 2P$^6$ 1G$^{18}$ 2D$^{10}$ 1H$^{22}$ 3S$^2$ 2F$^{14}$ 1I$^{26}$ 3P$^6$ 1J$^{30}$ 2G$^{18}$, and weaker energy gaps at 196 and 198 electrons, after completion of 3D$^{10}$ and 4S$^2$ shells, respectively. However, the calculated projected local density of

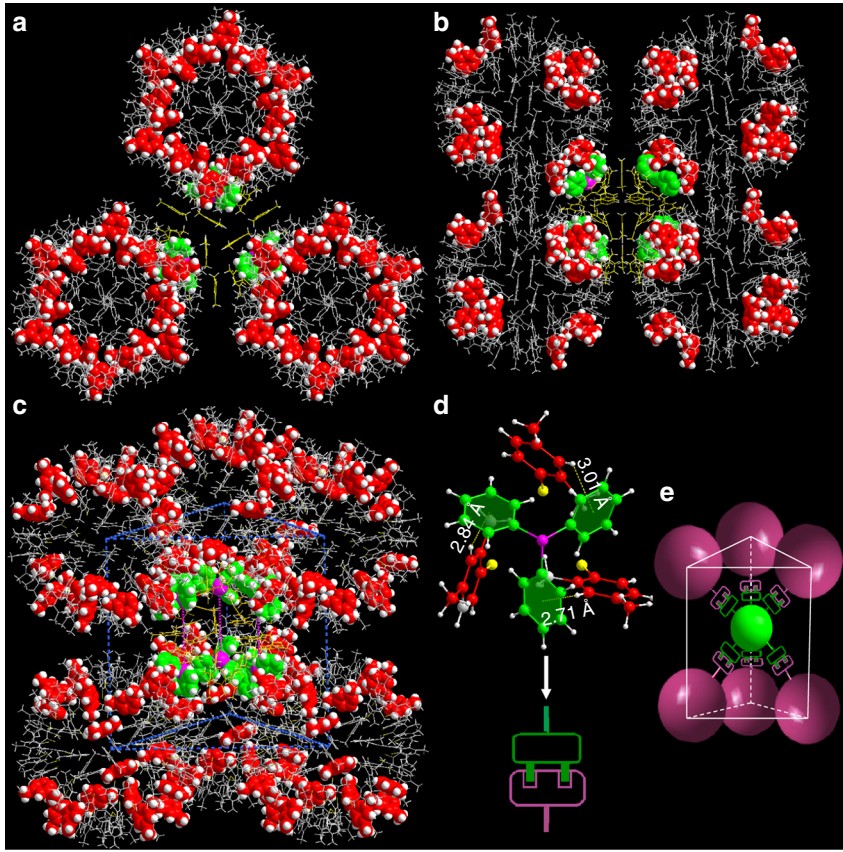

**Fig. 3** Interparticle self-assembly directed by the ligand anisotropy of surface patterns. **a–c** Arrangement of surface ligands of both (AuAg)$_{267}$ and (AuAg)$_{45}$ in the crystal lattice: top view (**a**) and side view (**b**) and (**c**). **d** The C-H···π interactions among the interparticle ligands, similar to tightly jointed plugs and sockets. **e** Scheme showing directional packing of binary composites via anisotropic ligand surface. Color code: magenta P, gray, red, gold, and green C, white H

electron states (PLDOS) of the three considered models for the (AuAg)$_{267}$ cluster do not show an energy gap at 186 electrons, rather there are two well-defined gaps at 190 electrons (0.10–0.16 eV) and 196 electrons (0.14–0.20 eV) (Fig. 5). The analysis of the PLDOS data of (AuAg)$_{267}$ clusters shows two deviations from the ideal shell order from the spherical jellium model: (i) both 3D$^{10}$ and 4S$^2$ shells are lowered in energy below HOMO, i.e., have become fully occupied, and at the same time (ii) the 1 J shell is widely broadened and fragmented on both sides of the HOMO energy, i.e., several 1J-symmetric states that are fully occupied in the spherical jellium model are now shifted upwards above the HOMO and are unoccupied. This re-organization of the shell structure is likely due to the lowering of symmetry from the perfect sphere by the discrete point-group symmetry of the core, as well as of the ligand layer. The lack of a detectable HOMO-LUMO energy gap demonstrates that the electronic structure does not provide a leading mechanism to stabilize the observed atomic structure of (AuAg)$_{267}$, hence, the stability must arise from the geometrical concentric packing of metal atoms to magic Mackay/anti-Mackay icosahedral shells and the structure of the protecting ligand shell, as discussed in detail above.

In contrast, the analysis of the electronic structure of the smaller (AuAg)$_{45}$ cluster (Fig. 5) shows a clear HOMO-LUMO energy gap (0.71 eV) as expected for 18 jelliumatic electrons (45–27 = 18). The projection to angular momentum components indicates the closing of the 1D$^{10}$ shell at 18 electrons, as expected, and another major energy gap (0.65 eV) at 20 electrons, i.e., after closing the 2S$^2$ shell. This cluster is thus clearly energetically stabilized by the electron shell closure at 18 electrons, i.e., at a magic electron number. The calculated UV-vis absorption

spectrum (Supplementary Fig. 13) agrees qualitatively with the measured one, showing a maximum at around 475 nm and a shoulder around 580 nm, slightly red-shifted from the experimental values of 434 nm and 570 nm (Fig. 4a, inset), respectively.

**Electrochemical properties.** A typical differential pulse voltammetric (DPV) response for the as-prepared (AuAg)$_{267}$·(AuAg)$_{45}$ co-crystal dissolved in 0.1 M Bu$_4$NPF$_6$/CH$_2$Cl$_2$ at room temperature, exhibiting 14 evenly spaced peaks ($\Delta V$ is 0.16 eV on average) characteristic of charge injection to the metal cores is given in Fig. 4b, which is a clear confirmation that (AuAg)$_{267}$· (AuAg)$_{45}$ are indeed multivalent redox species. We did not observe a clear HOMO-LUMO gap among quantized double layer charging (QDL) reduction and oxidation peaks. Our DFT calculations support this observation, given that it is likely that the larger nanoparticle dominates the redox behavior. As shown in Supplementary Fig. 14, the dense spacing of electronic states of (AuAg)$_{267}$ around the HOMO yields a quadratic behavior of the total energy of the cluster $E$ as a function of charge $Q$, $E = aQ^2 + bQ + c$, akin to the charging of a classic metallic sphere. From the parabolic fits (Supplementary Fig. 14), a constant charging energy ($d^2E/dQ^2 = 2a$) can be evaluated according to the standard theory of metallic quantum dots[37]. The calculated constant charging energy (in vacuum) for the three considered models of (AuAg)$_{267}$ is about 1.02 eV per reduction/oxidation step, which indicates that the effective dielectric constant of the electrolyte is about 6.4, estimated as the ratio between the calculated (in vacuum) and measured (in solution) charging energy. The case of (AuAg)$_{45}$ is interesting, as the HOMO-LUMO energy gap $E_{HL} \cong 0.7$ eV and

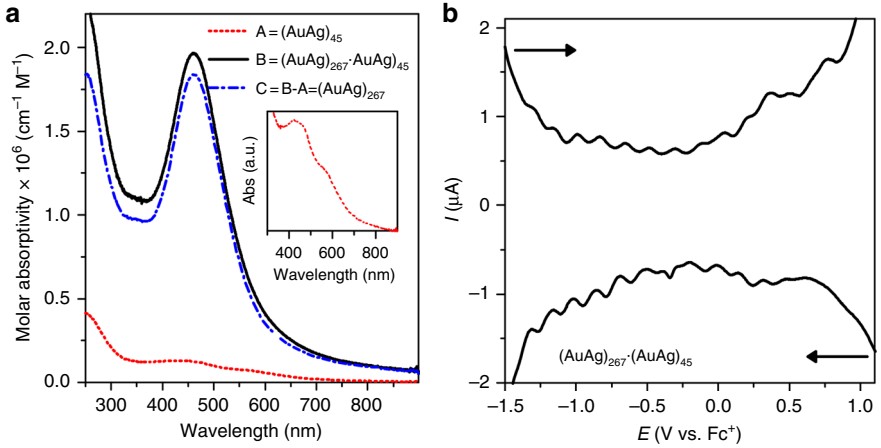

**Fig. 4** Optical and electrochemical properties of $(AuAg)_{267}\bullet(AuAg)_{45}$. **a** Molar absorptivity spectra of $(AuAg)_{267}\bullet(AuAg)_{45}$ cocrystal, plasmonic $(AuAg)_{267}$, and molecule-like $(AuAg)_{45}$ clusters dissolved in $CH_2Cl_2$. Inset plot is the enlarged view of UV-Vis spectra of $(AuAg)_{45}$. **b** DPV curves of $(AuAg)_{267}\bullet(AuAg)_{45}$ cocrystal dissolved in $CH_2Cl_2$

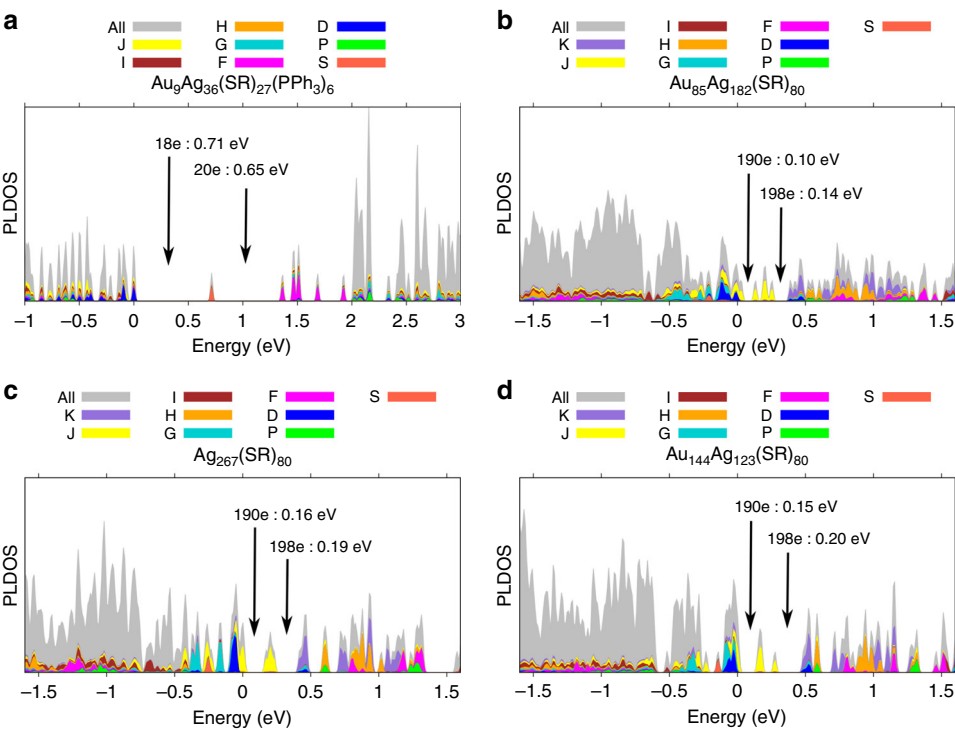

**Fig. 5** Calculated electronic structures of $(AuAg)_{267}$ and $(AuAg)_{45}$. **a** Projected Local Density of States (PLDOS) of the $(AuAg)_{45}$ cluster. **b–d** The same analysis for a hypothetical all-silver $Ag_{267}$ and for two intermetallic clusters with compositions of $Au_{85}Ag_{182}$ and $Au_{144}Ag_{123}$, all computed in the nanoparticle observed crystal structure of $(AuAg)_{267}$. The various colors denote the weights of orbitals projected onto spherical harmonics, centered at the cluster center-of-mass. The energy of the highest occupied molecular orbital (HOMO) is at zero. The major energy gaps around or above the HOMO energy are indicated by arrows

the cumulative gap at 20 shell electrons are both visible in the calculation of $E(Q)$, see Supplementary Figs. 14 and 15. Hence, the calculations predict that $(AuAg)_{45}$ should contribute just a few discrete charging peaks in the energy (voltage) region probed by the DPV experiment. The exact locations of those charging peaks are undetermined since the relative redox potentials of the larger and smaller particles are not known. However, we note that the experimental DPV data of $(AuAg)_{267}\cdot(AuAg)_{45}$ indeed is more complex than the data from $Ag_{206}$ that we reported recently[38], which may result from the presence of two particles having distinctly different redox behaviors.

In conclusion, the co-crystallization of a giant $(AuAg)_{267}$ nanoparticle and a much smaller $(AuAg)_{45}(2,4-SPhMe_2)_{27}$ cluster into multicomponent PtB-type hierarchical structures has been discovered. The present report exemplifies a bimodal particle nucleation and growth from solution driven by geometrically closed-shell and electronically closed-shell effects. Specifically, the plasmonic $(AuAg)_{267}$ nanoparticle adopts a magic shell packing, that is, a four concentric core-shell structure of the $Ag@M_{12}@M_{42}@M_{92}@Ag_{120}(SR)_{80}$ (M = Au or Ag) with an idealized $I_h$ symmetry, while molecular $(AuAg)_{45}$ cluster satisfies an 18-electron superatom shell closure. The 147-metal-atom solid

core in (AuA)$_{267}$ possesses a magic number of atoms, which was unprecedented in metal nanocluster family. Meanwhile, a well-organized metal segregation phenomenon, alternating Ag- and Au-rich shells behavior, was observed in the (AuAg)$_{267}$ metal framework. Besides, we analyzed the packing constraint of the (AuAg)$_{267}$·(AuAg)$_{45}$ co-crystal and its relation to surface ligand structures, which was helpful for the understanding of clustering process in unconventional PtB-structure superlattices.

## Methods

**Synthesis of the mixture of (AuAg)$_{267}$ and (AuAg)$_{45}$ (1)**. In a typical preparation, 10 mg of AgNO$_3$ or 20 mg of AgSbF$_6$ was dissolved in 1 ml of methanol, followed by the addition of 12 mg AuPPh$_3$Cl in 4 ml of dichloromethane. The mixture was cooled to 0 °C in an ice bath, 5 μL of 2,4-dimethylbenzenethiol, 4 mg PPh$_3$, and 10 mg of tetraphenylphosphonium bromide were then added. After 20 min of stirring, 1 ml of an aqueous solution of NaBH$_4$ (40 mg/mL) and 50 μl of triethylamine were added quickly to the reaction mixture under vigorous stirring. The reaction mixture was aged for 12 h at 0 °C. The aqueous phase was then removed. The organic phase was washed several times with water and evaporated for further analysis. Dark single crystals suitable for X-ray diffraction study were grown by a double-layer of hexane/CH$_2$Cl$_2$ solution of crude product at 4 °C for two weeks (Supplementary Table 1). The yield of (AuAg)$_{267}$·(AuAg)$_{45}$ was ~15% (based on Au).

**Synthesis of (AuAg)$_{45}$ nanocluster**. Same as (1) with the exception of increasing the usage of PPh$_3$ to 8 mg. Both (AuAg)$_{267}$·(AuAg)$_{45}$ (hexagonal-prismatic shape) and (AuAg)$_{45}$ (semi-thick hexagonal plate, main product and Supplementary Table 2) crystals are obtained (Supplementary Fig. 2). The yield of (AuAg)$_{45}$ was ~10% (based on Au).

**DFT calculations**. All the atomistic DFT computations were performed using the real-space code package GPAW[39]. The experimental crystal structures of (AuAg)$_{45}$ and (AuAg)$_{267}$ were used as such for further analysis of the electronic density of states and their projections to spherical harmonics. Three compositional models were built for the larger particle, one having a hypothetical all-silver Ag$_{267}$ core and two intermetallic cores Au$_{85}$Ag$_{182}$ and Au$_{144}$Ag$_{123}$. The compositions for the intermetallic clusters were selected to represent the absolute maxima and minima in integrated Au occupation, based on the probabilities (partial occupation numbers) of these metals in the refined crystal structure data. To study the charging behavior, the total energy of the clusters was evaluated as a function of charge via single-point calculations, i.e., ignoring the structural relaxation. The electron–electron interactions were described by the PBE-functional and the PAW setups for silver and gold included relativistic effects. The real-space grid spacing was 0.25 Å. Optical spectra were calculated by using the linear responsetime-dependent DFT module implementedin the GPAW software.

**Data availability**. The X-ray crystallographic coordinates for structures reported in this work (see Supplementary Tables 1 and 2) have been deposited at the Cambridge Crystallographic Data Centre, under deposition number CCDC-1839941 and 1839942. These data can be obtained free of charge from the Cambridge Crystallographic Data Centre via http://www.ccdc.cam.ac.uk/data_request/cif. All other relevant data are available from the corresponding authors.

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

## Acknowledgements

We thank the National Key R&D Program of China (2017YFA0207302) and National Natural Science Foundation of China (21731005, 21420102001, 21333008, and 21721001) for financial support. The financial support (to B.K.T.) from *i*ChEM, Xiamen University is gratefully acknowledged. H.H. acknowledges the support from China's National Innovation and Intelligence Introduction Base visitor program. The work in University of Jyväskylä was supported by the Academy of Finland (grants 266492, 294217, and H.H. Academy Professorship). The computations were done at the CSC −the Finnish IT Center for Science in Espoo (Finland) and in Barcelona (Spain) supercomputing center under PRACE project "Nanometals".

## Author contributions

N.F.Z. designed the study, supervised the project, and analyzed data. J.Z.Y. conceived and carried out experiments and analyzed the data. B.D. assisted the X-ray structure analysis. C.Y.H. and J.P. carried out the measurements of TEM and EDS. S.M. and H.H. performed the density functional theory computations and analyses. N.F.Z., H.H., B.K.T., B.D., L.S.Z., and J.Z.Y. wrote the manuscript. All the authors discussed the results and contributed to the preparation of the manuscript.

## Additional information

**Competing interests:** The authors declare no competing interests.

