## [Peer Review File · Nature Communications]

Reviewers' comments:

Reviewer #1 (Remarks to the Author):

This paper reports a joint experimental and theoretical study of atomically precise synthesis of ligand protected metal nanoclusters. The important result of this paper is that the authors have discovered co-crystallization of two nano-clusters composed of (AuAg)₄₅ and (AuAg)₂₆₇ core. The former is stabilized by electronic shell closure while the later is stabilized by atomic shell closure. One of the most important discoveries in cluster science has been the presence of magic numbers and their potential as budding blocks of a new class of materials - cluster assembled materials. Ways have been found to make this possible. This papers shows that in a single experiment one can observe both the electronic shell closure and atomic shell closure effect. The paper is well written, the results are important, and I recommend its publication in Nature Communications, subject to the following implementation. That cluster assembled materials could be synthesized with magic clusters as building blocks and that both electronic and atomic shell closings are important was first brought into focus by Khanna and Jena (Phys. Rev. Lett. 69, 1664 (1992); Phys. Rev. B 51, 13705 (1995)). These papers should be cited.

Reviewer #2 (Remarks to the Author):

The authors report

- (1) the synthesis of atomically pure Ag-Au metal-nanoclusters, a two-layered M₄₅ and a four-layer M₂₆₇ one, regularly protected basically by –thiolate ligands and also by some ←phosphine ligands ;
- (2) the preparation of 1:1 co-crystals ;
- (3) the full structure resolution of the co-crystal by single crystal X-ray diffraction, resolving the structural details, exhibiting the well described regularities ;
- (4) the UV-Vis absorption spectra, confirming the smaller molecular and the larger nano metallic character of the two units ;
- (5) the further characterization by differential pulse voltammetry ;
- (6) and finally the quantum-chemical PBE-DFT calculational reproduction and electronic analysis of the two units.

The report and discussion is clear and concise. It is a great paper.

I have only three questions:

(1) A small point: Concerning Fig. 4a, the red curve A is for the small unit in solution, the curve B for the co-crystal. What is curve C? According to the legend (which may be written more clearly) I would guess that C is for the large unit in solution – however in the Figure the wording suggests that C is the difference of co-crystal minus small-unit-solution spectra. Please clarify.

(2) Another minor point: How well do the experimentally derived and the theoretically optimized geometries agree with each other? Please add a short sentence concerning typical deviations and trends.

(3) Did the authors account for the seriously relativistic behaviour of the metallic (gold) valence electrons? I could not find any word about that. If a nonrelativistic model was applied, the calculations are somewhat meaningless, and the agreement with reality must be due to large error cancellation. Then we MUST think what to do.

Reviewer #3 (Remarks to the Author):

The authors present a fascinating discovery of a co-crystal of two different sizes of noble metal nanoparticles. It is the first observation to my knowledge of a co-crystal of two differently-sized clusters. It is hard to purposefully create co-crystals yet they have the potential for novel properties, so anything that can be learned about this type of system is very valuable. The individual nanoparticles themselves also have unique and highly interesting structures. I recommend publication in Nature Communications after the following minor points are considered.

I have only minor comments about the manuscript:

p.4 rephrase "or by simulated based on energy consideration "

p.4: this doesn't make sense to me: " $x=93.8 \pm 0.1$ for Au_xAg_{1-x} " with this formula, I would expect x between 0 and 1. it doesn't seem to fit with a 2.33:1 ratio. should subscript 1 be 312 instead?

p.5: is the wrong figure listed in the following statement "the 9 atoms in M9 tricapped trigonal prism core of the $(AuAg)_{45}$ cluster, are simply composed with occupancies exceeding 95% (see Supplementary Fig. 7)." ? the figure is for 267, not 45.

p.6: how were the nanoparticles separated in order to get the individual components of the absorption in Fig. 4?

p.7: spelling "1J-symmteric"

Caption of Figure 1: rephrase "The individual structures of the plasmonic $(AuAg)_{267}$ nanoparticle co-crystallized and of the molecule-like $(AuAg)_{45}$ cluster. " it is called "tri-prism" in caption, but is normally referred to as trigonal prism

Figure 2 caption: forth should be fourth; meaning of "severally capped" is unclear

The authors could mention something about the single crystal $[(AuAg)_{45}]$ results in the SI - a picture would be nice to show its differences from the co-crystal figures.

Reply to Referee # 1

We appreciate the helpful comments from the reviewer. According to the reviewer's suggestions, we have carefully revised our manuscript as follows:

1. **Comment:** This paper reports a joint experimental and theoretical study of atomically precise synthesis of ligand protected metal nanoclusters. The important result of this paper is that the authors have discovered co-crystallization of two nano-clusters composed of $(\text{AuAg})_{45}$ and $(\text{AuAg})_{267}$ core. The former is stabilized by electronic shell closure while the later is stabilized by atomic shell closure. One of the most important discoveries in cluster science has been the presence of magic numbers and their potential as budding blocks of a new class of materials - cluster assembled materials. Ways have been found to make this possible. This papers shows that in a single experiment one can observe both the electronic shell closure and atomic shell closure effect. The paper is well written, the results are important, and I recommend its publication in Nature Communications, subject to the following implementation. That cluster assembled materials could be synthesized with magic clusters as building blocks and that both electronic and atomic shell closings are important was first brought into focus by Khanna and Jena (Phys. Rev. Lett. 69, 1664 (1992); Phys. Rev. B 51, 13705 (1995)). These papers should be cited.

Response: Thank the reviewer for bringing the references to our attention. We have cited these two references in our revised manuscript as Ref. 11 and 12.

Reply to Referee # 2:

Thank the reviewer for the publication recommendation and also helpful comments.

1. **Comment:** A small point: Concerning Fig. 4a, the red curve A is for the small unit in solution, the curve B for the co-crystal. What is curve C? According to the legend (which may be written more clearly) I would guess that C is for the large unit in solution – however in the Figure the wording suggests that C is the difference of co-crystal minus small-unit-solution spectra. Please clarify.

Response: Thank for the reviewer's carefully attentions. We have added curve $C=B-A=(\text{AuAg})_{267}$ in Fig. 4a. And yes, the curve C is for the large $(\text{AuAg})_{267}$ unit in solution. Since the pure $(\text{AuAg})_{267}$ unit have not been obtained yet, we just indirectly calculated its molar absorptivity via the difference of co-crystal minus small-unit-solution spectra.

2. **Comment:** Another minor point: How well do the experimentally derived and the theoretically optimized geometries agree with each other? Please add a short sentence concerning typical deviations and trends.

Response: All the calculations for the large and small clusters were done with an atomic configuration fixed to the experimental structures.

3. **Comment:** Did the authors account for the seriously relativistic behaviour of the metallic (gold) valence electrons? I could not find any word about that. If a nonrelativistic model was applied, the calculations are somewhat meaningless, and the agreement with reality must be due to large error cancellation. Then we MUST think what to do.

Response: We thank the referee for asking about inclusion of the relativistic effects. We do it routinely for all heavy metals. We have added a clarification of this into the methods description

Reply to Referee # 3

We thank the reviewer for the helpful comments. According to the suggestions, we have carefully revised our manuscript as follows:

1. **Comment:** p.4 rephrase “or by simulated based on energy consideration”

Response: As suggested, we redrafted “or theoretically forecasted” in new version.

2. **Comment:** p.4 this doesn't make sense to me: “ $x=93.8\pm 0.1$ for $\text{Au}_x\text{Ag}_{312-x}$ ” with this formula, I would expect x between 0 and 1. it doesn't seem to fit with a 2.33:1 ratio. should subscript 1 be 312 instead?

Response: Thanks for pointing out this mistake. We have changed “ $x=93.8\pm 0.1$ for $\text{Au}_x\text{Ag}_{1-x}$ ” to “ $x=93.8\pm 0.1$ for $\text{Au}_x\text{Ag}_{312-x}$ ”.

3. **Comment:** p.5: is the wrong figure listed in the following statement “the 9 atoms in M_9 tricapped trigonal prism core of the $(\text{AuAg})_{45}$ cluster, are simply composed with occupancies exceeding 95% (see Supplementary Fig. 7).”? the figure is for 267, not 45.

Response: Thanks for pointing out this mistake. We have modified this sentence to “the 9 atoms in M_9 tricapped trigonal prism core of the $(\text{AuAg})_{45}$ cluster, are simply composed Au with occupancies nearly 100% [see the cif file for experimental atomic configuration of $(\text{AuAg})_{45}$].”

4. **Comment:** p.6: how were the nanoparticles separated in order to get the individual components of the absorption in Fig. 4?

Response: In fact, we were unable to get the pure $(\text{AuAg})_{267}$ unit yet. Since the absorptivity of cocrystal are contributed by only two components, large $(\text{AuAg})_{267}$ and small $(\text{AuAg})_{45}$, it was reasonable to consider the molar absorptivity of $(\text{AuAg})_{267}$ nanoparticle was equivalent to difference of co-crystal minus small-unit-solution spectra based on Lambert-Beer's Law.

5. **Comment:** p.7: spelling “1J-symmteric”

Response: We changed “1J-symmteric” to “1J-symmetric”.

6. **Comment:** Caption of Figure 1: rephrase “The individual structures of the plasmonic $(\text{AuAg})_{267}$ nanoparticle co- crystallized and of the molecule-like $(\text{AuAg})_{45}$ cluster.” it is called “tri-prism” in caption, but is normally referred to as trigonal prism

Figure 2 caption: forth should be fourth; meaning of “severally capped” is unclear

The authors could mention something about the single crystal $[(\text{AuAg})_{45}]$ results in the SI - a picture would be nice to show its differences from the co-crystal figures.

Response: As suggested, we have revised these mistakes and added the packing structure of the individual $(\text{AuAg})_{45}$ nanocluster (Supplementary Fig. 7) to show its differences from the co-crystal figures.